# Morphology, Physiology and Analysis Techniques of Grapevine Bud Fruitfulness: A Review

Ana I. Monteiro *, Aureliano C. Malheiro and Eunice A. Bacelar

Centre for the Research and Technology of Agro-Environmental and Biological Sciences (CITAB), University of Trás-os-Montes e Alto Douro, 5000-801 Vila Real, Portugal; amalheir@utad.pt (A.C.M.); areale@utad.pt (E.A.B.)
* Correspondence: anamonteiro@utad.pt

**Abstract:** Grapevine reproductive development extends over two growing seasons (vegetative cycles), for the complete formation of inflorescences and clusters. Induction and floral differentiation, the mechanism that leads to the formation of reproductive structures inside dormant buds, is a complex process divided into three well-defined stages (formation of anlagen, inflorescence primordia and flowers). This sequence of stages comprises morphological, biochemical, and physiological events, influenced by a set of environmental and endogenous factors. Inflorescence primordia formation determines the potential number of clusters that will be formed in the following growing season. Thus, during bud dormancy, viticulturists and winemakers can obtain a first yield prediction through the determination of bud fruitfulness. This information allows adjustments to be made to bud load, promoting balanced yield and fruit quality and higher commercial value. The present review describes the morphology and physiology of the formation of inflorescence primordia, as well as discusses the main abiotic and biotic factors involved, including a physiological disorder known as primary bud necrosis. In the same way, we intend to approach the more used techniques of analysis of fruitfulness and its importance for a robust yield forecasting.

**Keywords:** bud differentiation; inflorescence primordia; potential yield; reproductive structures; *Vitis vinifera*



## 1. Introduction

Grapevine (*Vitis vinifera* L. and interspecific hybrids) is one of the most economically important fruit crops in the world. The sector's sustainability, mostly devoted to wine production (57%) [1], is greatly influenced by fluctuations in annual harvests. These yield variations, with a high impact on berry quality, are frequently due to changes in the number of inflorescences formed per grapevine, in the number of flowers per inflorescence, and in the fruit set and fruit weight [2–4]. Environmental conditions (e.g., air temperature), diseases, pests, and cultural practices (e.g., winter pruning) also have an important effect [4,5].

Winter pruning is a first viticultural practice through which yield can be regulated and quality improved [6]. Each year, during dormancy, the bud load is adjusted according to the bud fruitfulness in order to meet the productive objectives [7]. However, and remarkably, grapevine reproductive development extends over two vegetative cycles (growing seasons) (Figure 1). It begins with inflorescence primordia formation in first year and with differentiation of the flowers, development of the clusters until the physiological maturation of berry and seeds in following year [8].

Determined by the differentiation of anlagen in inflorescences during the first vegetative cycle, bud fruitfulness represents the first measure of productive potential, as it defines the number of bunches that will be formed [5,9–11]. Thus, bud fruitfulness provides an estimate of the potential yield for the following season [5,12]. It depends on the variety, type of bud and position of the bud along the shoot whose effect manifests in terms of the number of inflorescences per bud and size (number of flowers) [8,13]. Bud fruitfulness is

generally lower in basal buds and gradually increases until reaching maximum values in the 3rd and 4th bud, then decreasing [14,15]. Buds are considered fruitful when they have at least one primordium inflorescence. Conversely, the bud is considered infertile in the absence of inflorescence primordia or the existence of only tendril primordia and leaves [16].

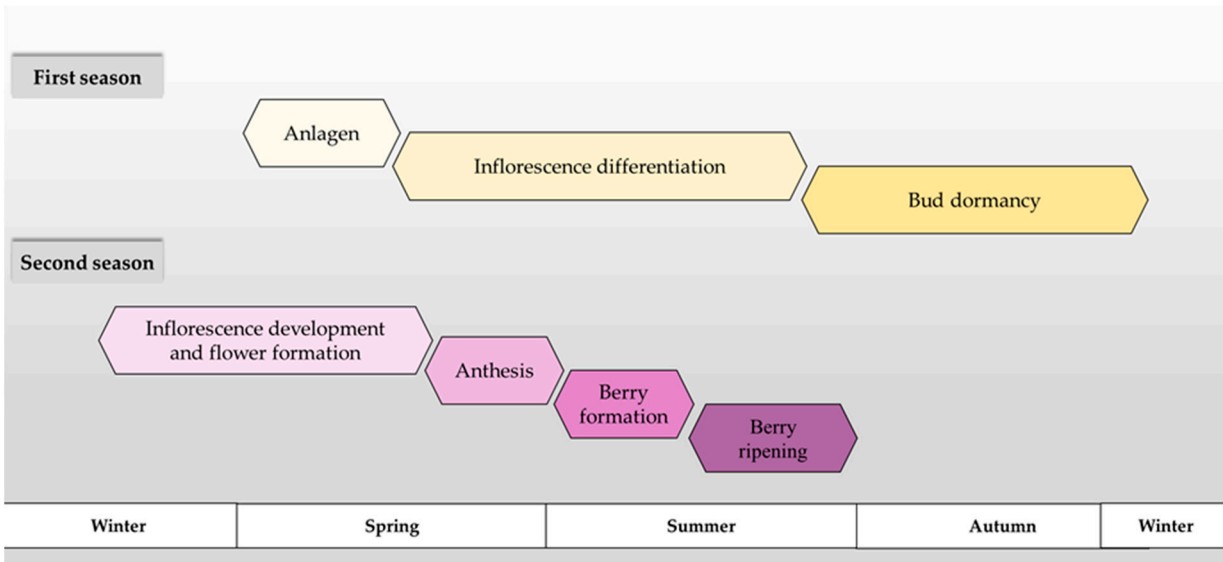

**Figure 1.** Grapevine reproductive cycle showing the sequence of cluster formation events over two growing seasons.

Therefore, the main objective of the present work is to provide an integrated overview of the morphology and physiology of axillary meristems and buds, as well as the effects of abiotic and biotic factors on the mechanisms of induction and floral differentiation. In addition, the most used techniques of fruitfulness analysis are discussed.

## 2. Grapevine Buds: Morphology, Structure and Function

The development and the morphology of the grapevine buds have been described previously [17–21]. During shoot development, an axillary bud complex can potentially develop at the axil of a leaf (base of the petiole) [15,22]. In normal ontogeny, buds undergo dormancy (dormant bud), except when they develop in the same season as they are differentiated, in which case they produce a sylleptic shoot that can be fruitful, but rarely reaches comparable quality of production [8,23].

The dormant bud remains in a state of dormancy until the following year due to hormonal inhibition of the apex of lateral shoots [8,24]. Anatomically, the dormant buds comprise a larger central bud, which corresponds to a primary bud and two smaller buds (secondary and tertiary buds) on either side of the primary. Due to their complex structure, dormant buds are also defined in the literature as compound buds. Figure 2 shows a transverse section of a dormant bud with a primary and two secondary buds protected by the bud scales. Generally, the primary bud develops into a new fruiting shoot in spring, while the secondary and tertiary buds remain dormant. If the primary bud is damaged or dies, the secondary bud may develop a shoot to compensate for the loss [25–28]. However, these buds have a lower fruitfulness than the primary. The secondary buds may form one or more inflorescence primordia in some varieties and tertiary buds do not usually produce inflorescences [24,29].

The formation of the primordia of all vegetative and reproductive organs occurs inside the dormant buds, being the whole structure well protected by a set of bracts and epidermal hairs (protection from unfavorable weather conditions, insect damage, or diseases) [8,20,30]. After the formation of reproductive structures, dormant buds go into dormancy until next spring when they restart their growth in response to environmental conditions, and complete their development with the formation of flowers and berries [15,16,21,30]. These

are the most fruitful buds, as they have undergone a long process of differentiation with higher energy needs [8]. They usually give rise to one or two inflorescences per bud, depending on variety, bud position in the shoot, and abiotic and biotic conditions.

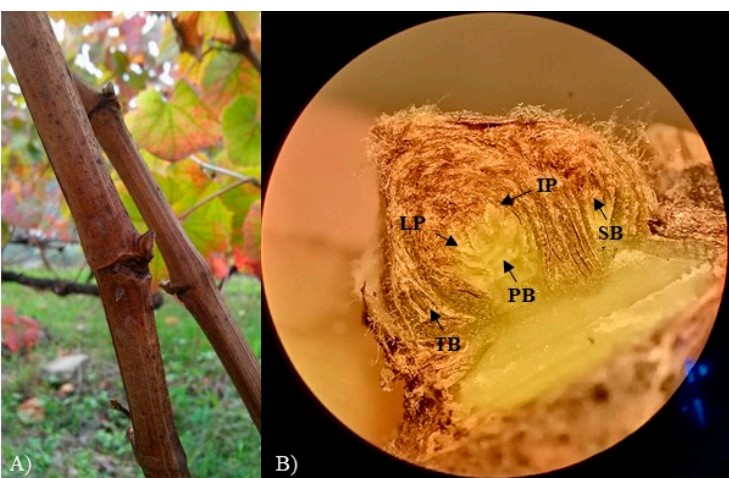

**Figure 2. (A)** Bud of Fernão-Pires variety during the dormant period surrounded in set of bracts that protects them from unfavorable environmental conditions, pests, and diseases. **(B)** Longitudinal section of a dormant bud of Loureiro variety obtained by dissection, showing bud position: in a central position the primary bud (PB) and the secondary (SB) and tertiary bud (TB) on each of primary bud. LP-Leaf primordia, IP-Inflorescence primordia.

## 3. Inflorescence Morphogenesis and the Appearance of Flower Primordia

The successive stages of floral initiation and differentiation have been described based on morphological and histological studies using electronic microscopy [29,31–34]. The formation of inflorescence and flowers is initiated in dormant buds and results from a long process that involves three main stages, which includes two growing seasons [29].

The first stage starts with uncommitted primordia (UP) formation that is also designed by anlage or anlagen [11,15,16,35,36]. Anlagen arises like club-shaped meristematic protuberance from the apices of the primary buds and it represents the first step of inflorescences formation [19,29,36,37]. The anlagen will differentiate and can generate inflorescence primordia or tendril primordia or, sometimes, an intermediate structure, depending on environmental conditions and hormonal factors [11,16,29,37,38]. Then, in the second stage, the development of anlagen starts with the formation of bract and splits into two unequal parts; inner and outer arm. Both arms have the potential to produce inflorescence primordia or tendril primordia [4,16,22]. The inflorescence primordia differentiation occurs by extensive branching of the anlagen [19,29]. The inner arm, the adaxial portion nearer to the apex, will have more potential to divide and form globular branch primordia, which will give rise to the main body of the inflorescence (rachis). The branching degree of the inner arm gradually decreases in an acropetal direction giving the inflorescence primordium a conical shape [19,22].

The outer arm, the smaller abaxial part adjoining the bract, will give rise to either a wing or a larger branch on the top of the cluster [29]. The branching of the outer arm is less extensive and develops into the lowest branch of the inflorescence. The differentiated inflorescence primordia are then formed by an axis with small protuberances in the axilla, where are inserted the globular rudimentary branches and the future flowers to be formed, resembling a cluster [8,39]. Both stages, crucial in bud fruitfulness, mark the beginning of floral initiation and establish the potential number of inflorescences and subsequently the number of clusters per bud that will be formed in the following season [21].

Flower differentiation is the third and last stage. After budburst, in the next spring, the inflorescence primordia continue differentiation to form floral organs on individual floral buds along the rachis [20]. Each branch of the inflorescence primordia divides successively

and finally forms the flower initials [37,38]. The appearance of the calyx as a continuous ring of tissue on the rim of the primordium marks the beginning of the flowering. The calyx comprises a continuous ring of tissue, which covers the whole flower primordium and forms an incomplete cap [19,29]. The petals, which develop at the same time as the calyx, become lobed and make their way through the calyx cap. As each petal elongates, cells are formed on its margins, which interlock with similar cells on the margins of the adjacent petals to form the calyptra [19,20,37,40]. Finally, the flower is fully developed and ready for anthesis. The arrangement of flowers on an inflorescence becomes visually clearer when the inflorescence begins to rapidly elongate before anthesis [6,8].

## 4. Tendril Primordia Formation

Tendrils and inflorescence primordia derive from similar meristematic structure (homologous structures), the anlagen or uncommitted primordium, but later on, they follow a divergent developmental pathway. When anlagen undergoes repeated branching, it gives rise to inflorescences, while producing few branches it turns into tendrils [16,34,37]. However, the anlagen differentiation is a complex mechanism, regulated by a set of stimuli of environmental and endogenous natures. A small imbalance in the factors involved during anlagen differentiation can cause it to differentiate into tendrils instead of differentiating in inflorescence primordia [16,20,38,41].

## 5. Factors Affecting Induction and Flower Formation

Different studies have focused on the environmental (abiotic) and endogenous (biotic) factors that directly and indirectly influence the process of induction and differentiation of inflorescence primordia [4,14–16,21,42]. Temperature, light, water status, and macronutrients availability are the environmental factors that most influence these processes. In addition, endogenous factors such as carbohydrate reserves (source/sink regulation), hormonal balance, and genetics also have an important role [4,15,21]. Thus, positive stimuli during the differentiation of anlagen will promote the inflorescence primordia development and have decisive impacts on the fruitfulness.

### 5.1. Environmental Factors

5.1.1. Temperature

A strong relationship between relatively high temperatures during the differentiation of inflorescence primordia and the number of inflorescences formed in dormant buds has been reported [29,42–45], although varietal dependent [4,46]. Studies carried out in a temperature-controlled environment with different varieties (Almeria, Muscat of Alexandria, Riesling, Syrah, and Thompson Seedless) have shown that the optimum temperature interval for the formation of inflorescences varies from 20 to 35 °C [47,48]. Temperatures below 20 °C enable tendrils formation, reducing bud fruitfulness and yield. On the other hand, temperatures above 30 °C, for at least 4 a 5 h per day, appear to be sufficient to induce the maximum number of inflorescence [16,47–49]. Buttrose [42] concluded that the three weeks before the anlagen formation is the critical period to high temperatures.

In addition, high temperatures are fundamental for the differentiation of the second and third inflorescences in many varieties, including cool climate varieties [16]. More recently, Watt, et al. [50] compared the time and extent of initiation and differentiation in Chardonnay primary buds in cool and hot climates (showed more advanced development).

However, the mechanism by which temperature affects the initiation of inflorescences remains unclear, although some hypotheses have been presented. Among the explanations is the influence of temperature on the biosynthesis of gibberellins (GAs) and cytokinins (CKs). Temperatures ranging from 25 to 35 °C promote CKs biosynthesis known to stimulate the differentiation of inflorescences [24]. Low temperatures (<20 °C) promotes GAs biosynthesis, responsible for promoting vegetative growth and limit nutrient accumulation. Another hypothesis is the influence of temperature on biochemical processes, particularly in photosynthesis, enzymatic and respiratory activities [27,51]. The optimal temperature

range for photosynthesis varies between 25 and 30 °C, depending on variety, phenology, and pedoclimatic conditions. Temperatures above 35 °C negatively affect the photosynthesis, as the stomata begin to close [27,52].

### 5.1.2. Light

Light (irradiance) has been described as a key factor for inflorescence initiation and development in dormant buds [5,15,17,44,47–49,53]. Buttrose [47,48] found that the number of inflorescence primordia increased with light intensity under controlled environmental conditions (growth cabinets with different light levels and photoperiod). These results are in agreement with other findings [44,46].

Under field conditions, the exposure of shoots to solar radiation increased the number of inflorescence primordia in Cabernet Sauvignon, Chardonnay, Flame Seedless, and Thompson Seedless varieties [46]. Conversely, low light intensity or total shading during initiation and differentiation can reduce the number and size of inflorescence primordia [16,53,54]. May and Antcliff [55] investigated the effects of shading on Thompson Seedless and concluded that a 70% reduction in light intensity for four weeks before anthesis drastically reduced bud fruitfulness. In addition, a reduction in number and size of inflorescence primordia occur in scion grafted onto vigorous rootstocks, that form dense canopies, subsequently, reduce the incidence of light inside canopies [44,56].

The importance of light in bud fruitfulness is essentially manifested by the influence of direct radiation on leaf photosynthetic activity and carbohydrate availability. Low light intensity reduces the amount of photoassimilates, limiting the carbohydrates supplied to the developing buds [5,15,27,36]. Therefore, it is essential to adjust canopy management practices, particularly trellis-training systems, shoot control, row spacing, and pruning intensity to ensure good exposure to radiation and avoid yield losses [5,8,57].

Although the photoperiod (day length) is not a particularly determining factor in inflorescence induction, it appears that the number of inflorescence primordia is greater on long days than on short days in some varieties [15,19]. As an illustration, Muscat of Alexandria showed to be a day-neutral variety while Riesling and Shiraz performed better on long days [47].

### 5.1.3. Water Status

The grapevine water status impacts the induction and differentiation of inflorescence primordia by the direct and indirect influence that water has on biochemical and biosynthetic processes, namely in maintaining cell turgidity, photosynthetic activity, and nutrient and photoassimilate transport [4,15,22,24,58,59]. The adequate water availability is reflected in improved conditions of differentiation of inflorescences contributing to an increase in bud fruitfulness [8]. Conversely, the number and size of inflorescences are negatively affected under water stress conditions. A study carried out in a controlled environment showed that the number and weight of Cabernet Sauvignon inflorescences decreased progressively with increased plant water stress [58]. Under water stress conditions, there is a reduction in the photosynthetic activity, and the carbohydrates produced may not be sufficient to provide the energy required for the differentiation of inflorescences [8]. On the other hand, a moderate water stress can increase bud fruitfulness due to reduced canopies density and improving bud light exposure, especially in the renewal zone [60].

Water stress also decreases the levels of CKs in xylem sap and increases abscisic acid (ABA) levels in leaves and stems, contributing to an imbalance in hormonal levels, with negative implications in the differentiation [15,19]. In response to the water stress, the increase of ABA endogenous levels induces stomata closure, inhibiting photosynthesis [59]. The differentiation of inflorescences is also sensitive to the combined effects of nitrogen deficiency and water stress [45].

### 5.1.4. Macronutrients Availability

An adequate supply of macronutrients (nitrogen, phosphorus and potassium) is particularly important to ensure optimum induction and floral differentiation and naturally increase bud fruitfulness [27,29]. Nitrogen (N) is a fundamental element in the composition of amino acids, which form structural proteins and enzymes, responsible for catalyzing all biochemical reactions. Furthermore, it is also an integral element of chlorophyll and hormones [27].

The number of inflorescence primordia and the number of flowers per inflorescence increased after N application to grapevines with low initial levels of N [19,45]. On the other hand, excessive N nutrition translates into an increase in plant vigor and increased shading of buds with negative effects for fruitfulness [15,61]. At the biochemical level, too much N promotes the biosynthesis of GAs [8].

Phosphorus is an indispensable macronutrient, as it is a structural constituent of biomolecules involved in energy metabolisms, such as nucleic acids and phospholipids [62]. Skinner and Matthews [63] found that the initiation, differentiation and maintenance of reproductive primordia were sensitive to the deficiency of this macronutrient.

Potassium (K) is involved in numerous physiological and biochemical processes, namely enzymatic activation, photosynthesis and plant water relations [64]. The application of K to the soil stimulated an increased Thompson Seedless fruitfulness [65]. In another study, K fertilization, during the first growth cycle, promoted an increase of 40–58% in the size of inflorescence primordia, depending on the bud position [66].

### 5.2. Endogenous Factors

### 5.2.1. Carbohydrate Reserves

The number of inflorescences developed depends on the availability of sugars and starch reserves accumulated in the perennial organs of the previous year and on the photosynthetic activity of the current year [36,67]. In order to better understand these relationships, different studies evaluated the effect of defoliation on the bud fruitfulness of the following year [35,68–70]. The authors concluded that early defoliation reduced root and trunk carbohydrate reserves associated with significant decreases (up to 50%) in the inflorescence number per shoot and flower number per inflorescence in the following season. The main cause for these findings may be the competition for photoassimilates between vegetative growth and induction and floral differentiation, which occur simultaneously [35,68]. On the other hand, leaf removal performed at a later stage, or under dense canopy conditions, can improve the exposure of the basal buds to light and, consequently benefit their fruitfulness [36,71].

### 5.2.2. Hormonal Balance

Induction and floral differentiation are processes mediated by the interaction between two hormones with opposed effects: GAs and CKs [15,16,22]. GAs synthesized at leaf level are responsible for the initiation of the anlagen, but later inhibit its development as an inflorescence, promoting the formation of tendrils [21,22,41]. Srinivasan and Mullins [16,37,41] observed that in certain genotypes it is possible to convert one form into another structure through the application of hormones or their inhibitors. Thus, exogenous application of GAs in form of gibberellic acid in young inflorescences converts them into tendrils or intermediate structures. On the other hand, the exogenous application of chlormequat (a gibberellin synthesis inhibitor) promoted inflorescence formation from anlagen and tendrils. This response can be explained by the inhibition of GAs synthesis and increases of endogenous CKs levels in the ascending sap [19,38].

Synthesized in the roots, CKs are transported via xylem sap to the target sites directly interfering in the differentiation of the inflorescence primordia and the different floral pieces [15,22,29,41]. Repeated applications of CKs at the apex of shoots showed that it can induce the formation of inflorescences in place of tendrils [16,37]. On the other hand, the application of synthetic CKs, PBA (6-(benzylamino)-9-(2-tetrahydropyranyl)-9H-purine) in tendril primordia isolated in vitro culture, induced the branching of these structures by

converting them into inflorescences [38]. Similarly, in twelve varieties of *Vitis vinifera* and six species of *Vitis* sp., the successive application of PBA at the shoot apex promoted the formation of inflorescences in place of tendrils. Thus, CKs stimulate the development of inflorescences from the lateral meristem, while GAs, although fundamental in the initiation of the primordia, inhibit the development of inflorescences, favoring the formation of tendrils [21,72]. In addition, since the hormones GAs and CKs can alter the rate of cell division in the anlagen, a "physiological time" can be defined as opposed to "chronological time. As an illustration, repeated pruning and N application (to stimulate vegetative growth) and GAs application followed by CKs, which also aimed to anlagen proliferation and shoot elongation, ended up shortening considerably time to fruitfulness of grape seedlings [73].

### 5.2.3. Genetic Factors

Induction and floral differentiation occur sequentially and under tight genetic control [40]. The analysis of the molecular regulatory network that controls the different stages of reproductive development is based on the identification and functional analysis of *V. vinifera* orthologous of *Arabidopsis thaliana* genes involved in flowering-signal integration, establishment of flower-meristem and flower-organ identity [20,30]. Molecular studies in *V. vinifera* showed that many genes expressed during flowering are also expressed in dormant buds at the time of the initiation and differentiation of inflorescence primordia. The ability to induce and form floral organs occurs through a complex network of nonlinear relationships between genes and their products (proteins, miRNAs). These relationships have been described in several species, especially in *A. thaliana* [74].

The flowering locus T (FT)/terminal flower 1 (TFL1) gene family has an important role as a flowering signal integrator. This gene family homologous to the *A. thaliana* encodes proteins with similarity to phosphatidylethanolamine binding proteins (PEBPs), which function as promoter and repressor of flowering [21,75–77]. Phylogenetic analyses identified at least five members of grapevine FT/TLF1 gene family that are grouped in three subfamilies: MFT-like, FT-like and TFL1-like. The VvFT gene (FT orthologue) is expressed in dormant buds and during the initial stages of inflorescence development [21,75]. Additionally, VvFT and VvMFT expression are associated with meristem determination and differentiation of organs, such as inflorescences, flowers or tendrils supporting the role of these genes as flowering promoters. On the other hand, VvTFL1A, VvTFL1B and VvTFL1C genes (that belong to TFL1-like family) are associated with vegetative development and maintenance of meristem indetermination within the bud [74,75,78,79].

Early flowering can be induced when certain genes are over-suppressive. Among the genes involved in the flowering process is the transcription factor known as VvVFL (AtLEAFY orthologue) expressed during the anlagen of dormant buds in the Riesling and Tempranillo varieties [21]. Some gene families are also involved in the initiation and differentiation of inflorescence primordia and tendrils in the Cabernet Sauvignon and Tempranillo varieties [72,80]. Among these genes, it was reported the VvSOC1 (homologous to SOC1 e *A. thaliana*) and VvMADS8 (a subfamily of the MADS-box gene of MADS transcription factors that are positively regulated during the first stages of inflorescence development and return to normal level in later stages of flowering development).

The most important hormones in the inflorescence initiation and differentiation process, GAs and CKs, act at the transcriptional level. Genetic evidence supports the inhibiting role of GAs in the differentiation of inflorescences, through the mutant phenotype of gibberellin-insensitive grapevines, with a mutation in the VvGAI gene, where all the tendril primordia differentiated in multiple inflorescences throughout the aerial part, even in the buds where there was less prevalence to form tendrils. This phenotype comes to support the hypothesis that GA-dependent signal-transduction pathway inhibit the differentiation of inflorescences [20,30,74].

Research studies based on transcriptional analyses during the development of inflorescences and tendrils, found that these two homologous organs initially share a common transcriptional program related to cell growth and proliferation functions [72]. In more

advanced development stages, they display specific genic expression programs related to the differentiation processes that occur in each of them. Tendrils have a higher transcription of genes related to photosynthesis, hormonal signaling and secondary metabolism, unlike inflorescences that have a higher transcription of genes encoding transcription factors, especially those belonging to the MADS-box family.

## 6. Analysis Techniques of Bud Fruitfulness

### 6.1. Bud Dissection and Histological Analysis

During dormancy, it is difficult to identify and quantify the fertility of buds, requiring the use of specific laboratory techniques and procedures, allowing to evaluate the inflorescence primordia [13,43]. The dissection and histological analysis of buds, in addition to helping to quantify the number of inflorescence primordia in each bud, allow to analyze the bud viability, namely to diagnose possible necrosis in the tissues and to evaluate the extent of the lesion in the tissue [26,81–83].

The dormant bud dissect method uses a stereomicroscope supported by tweezers and a scalpel blade to easily observe and identify the inflorescence primordia [13,17,43]. However, it is a time-consuming procedure, which requires a lot of care to avoid damaging the structure as the cuts and removal of protective structures are carried out. Small and careful cuts are made, from top to bottom towards the base or in the longitudinal direction to reveal the entire interior of the bud. The fragility of the primordia and the meristems and the difficulty in removing the epidermal hairs that line these structures and hinder visualization are the major problems. The great advantage of this technique is that the information is immediately available after the anatomical cut of the bud. In Australia, dissection for fruitfulness estimation is a widely used technique, with commercial laboratories providing this service, assisting winegrowers in making decisions on the winter pruning intensity [83].

In histological analysis, the buds are embedded in paraffin wax, sectioned, stained and observed under an optical microscope [26,43]. Similar to another histological technique, its success is related to the process of fixation plant material. After collecting the buds, the scales and epidemical hairs are removed to allow the fixatives solutions and paraffin wax to act and preserve the entire structure [43]. In order to increase the effectiveness of the practice, bud cuts must be made in series to guarantee the visualization of all the inflorescence primordia [13]. This is a time-consuming technique and requires equipment and reagents at higher costs [13,17,53,81].

Although with technically different procedures, all these methods need a good knowledge of the anatomy of the segment for unambiguous identification of all structures and primordia. A poor identification or non-visualization of inflorescence can lead the operator to mistakenly consider as infertile [13,43].

The inflorescence primordia fully developed are a small axis with many protuberances that will give place to the flowers (Figure 3). Usually, the inflorescence primordia are near apical meristem [39]. Microscopic analyses of buds are destructive methods. Therefore, the evaluated buds should not be destined for production, but those that would be suppressed in pruning [13,43]. This method can be inaccurate, especially for small buds, such as the basal buds, which are usually the most important for pruning [43].

These techniques are also useful to analyze the bud viability, assessing the existence of tissue damage, namely necrosis that can compromise its development [82]. The incidence of primary bud necrosis (PBN) is one of the causes of detrimental impacts on yield, due to the loss of the most fertile buds [15]. PBN is described as a physiological disorder resulting in the death of the primary bud during bud initiation [5,61,82,84,85]. Although the healthy secondary bud develops, but often less fruitful, to compensate for the loss of the primary one, lower yields are obtained [5,28,83]. Since the external appearance of a necrotic dormant bud is identical to a normal bud, its diagnosis and identification involve anatomical sections to observe internal structures, through bud dissection and histological observations [5]. Susceptibility to PBN also seems to be varietal dependent. A few studies

reported a high PBN incidence in Shiraz [61,82], Riesling [84], and Thompson Seedless (Sultana) varieties [86,87]. The PBN has been also associated with other factors, namely high shoot vigor [88], rootstocks of *American* species of *Vitis* [89], exogenous application of gibberellic acid [83,90], canopy shading [87], excessive irrigation [91] and low bud carbohydrates content [84,89]. Thus, it is important to make an early diagnosis of PBN, so that the number of buds left at winter pruning can be adjusted [5,28,83].

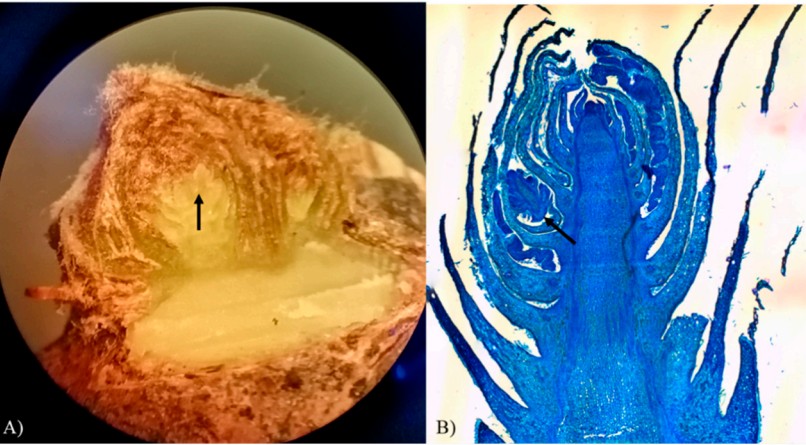

**Figure 3.** (**A**) Longitudinal section of bud tissue observed under a stereomicroscope at 20× magnification. (**B**) Longitudinal section of a dormant bud (Fernão-Pires variety) observed under a light microscope. Black arrow, in each panel, indicates inflorescence primordia.

### *6.2. Forcing Dormant Buds in Controlled Environmental Conditions*

Forced cultivation under controlled conditions also allows an early indication of the number of inflorescences developed in each bud [2,43,81,92]. Forcing buds presupposes the use of two-year-old wood containing one or more dormant buds, and subjecting them to controlled conditions of temperature, relative humidity, irradiance and photoperiod to induce the budburst [13,81,92]. Fruitfulness is determined by simple visual observation and counting the number of inflorescences in the young shoot. Forcing budburst of dormant buds is a simple and expeditious method, which does not require detailed knowledge about the anatomy of buds. However, the results are not immediate, as is necessary to wait for the development and visualization of inflorescences [2,81]. Although the plant material can be harvested at any time during the dormancy, in case the dormant buds have not accumulated enough chilling hours, the cuttings must be subjected to a chemical or physical agent to break the dormancy artificially. This process promotes the budburst to occur more quickly and homogeneously. In case the cuttings are collected close to budburst onset, the chilling requirements are already being fulfilled [43]. Low percentage budburst and fruitfulness of the base buds in some varieties represents a limitation to the use of this methodology [2].

### 7. Conclusions

The yield fluctuations verified annually are the result of a set of abiotic and biotic factors that influence the formation and development of grapevine reproductive organs. One of the starting points to minimize yield irregularities is prior knowledge of bud fruitfulness. During the dormancy, the dormant buds contain a specific number of all reproductive and vegetative organs that constitute the future shoots. At this phenological stage, identifying and quantifying the number of inflorescences allows knowing the number of clusters to be formed. This information permits the first advance yield estimation, which brings advantages mainly in the adjustment of the crop load at winter pruning. Although the identification of bud fruitfulness is not possible by direct observation during dormancy, methods of budburst forcing or laboratory techniques and procedures make it possible.

Even though early yield estimation based on bud fruitfulness is a useful tool for viticulturists and winegrowers, it does not take into account the detrimental impacts that

may occur later on the season, caused by factors such as severe weather conditions (e.g., hail damage), incidence of pests and diseases and eventually incorrect cultural practices.

**Author Contributions:** Conceptualization, methodology, validation, formal analysis, investigation and resources, all authors; writing—original draft preparation, A.I.M.; writing—review and editing, all authors; visualization and supervision, A.C.M. and E.A.B.; project administration and funding acquisition, A.C.M. All authors have read and agreed to the published version of the manuscript.

**Funding:** The study was funded by the INTERACT project–"Integrated Research Environment, Agro-Chain and Technology", no. NORTE-01-0145-FEDER-000017, in its line of research entitled VitalityWine, co-financed by the European Regional Development Fund (ERDF) through NORTE 2020 (North Regional Operational Program 2014/2020).

**Acknowledgments:** This work is supported by National Funds by FCT–Portuguese Foundation for Science and Technology, under the project UIDB/04033/2020. The first author acknowledges the grants' support with references BI/INTERACT/VW/412/2016 (INTERACT project), BIM/UTAD/56/2019 and BI/UTAD/21/2020 (project "To Chair–The optimal challenges in irrigation", PTDC/MAT-APL/28247/2017 with reference POCI-01-0145-FEDER-028247, funded by FCT).

**Conflicts of Interest:** The authors declare no conflict of interest. The funders had no role in the design of the study; in the collection, analyses, or interpretation of data; in the writing of the manuscript, or in the decision to publish the results.

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
