# Peer review of "Morphology, Physiology and Analysis Techniques of Grapevine Bud Fruitfulness: A Review"

_agriculture, doi:10.3390/agriculture11020127_

Round 1

Reviewer 1 Report

Lines 50 and 80: the buds in grapevine cannot be considered vegetative, but unproductive, since the tendrils (always present) are homologous structures of inflorescences (i.e. lines 138-140).

In all the text: latent buds are not synonyms of dormant (or quiescent) buds; they are the dormant buds that do not break the next season of formation and remain latent precisely, in the old wood, and possibly breaking later following particular conditions such as high vigor, wood damages …., giving the suckers. The term latent should be changed in dormant.

Author Response

We are thankful for the reviewer’s comments. The corresponding changes in the manuscript are highlighted in red font.

The term ‘latent’ was changed in dormant throughout the text.

Reviewer 2 Report

Review agriculture-1079650

This review paper describes the morphology and physiology of grapevine buds along with the methods for flower bud prediction. The contents on the morphology, structure, formation of the buds, inflorescence, and tendril have been well-known. In addition, environmental factors for inflorescence formation and the prediction method for buds were also well summarized in some review papers. It is, therefore, difficult to find incentive for publishing this review paper: I feel that something has come too late, but this review may have some merits for the viticulturists and winemakers, when more additional information would be provided. I ask the authors to add more information about the state-of-the-art information about the molecular basis mechanisms underlying grapevine bud initiation, differentiation, development, including the physiological function of FT and TFL etc.  

Author Response

Reply: We are grateful to the reviewer’s comments. The corresponding changes in the manuscript are highlighted in red font.

Additional information regarding ‘molecular basis mechanisms´ was added as follows:

Lines 301-313: “The Flowering Locus T (FT)/ Terminal Flower 1 (TFL1) genes family has an important role as a flowering signal integrator. This genes family homologous to the A. thaliana encodes proteins with similarity to phosphatidylethanolamine binding proteins (PEBPs), which function as promoter and repressor of flowering [21,74-76]. Phylogenetic analyses identified at least five members of grapevine FT/TLF1 gene family that are grouped in three subfamilies: MFT-like, FT-like and TFL1-like. The VvFT gene (FT orthologue) is expressed in dormant buds and during the initial stages of inflorescences development [21,74]. Additionally, VvFT and VvMFT expression are associated with meristem determination and differentiation of organs, such as inflorescences, flowers or tendrils supporting the role of these genes as flowering promoters. On the other hand, VvTFL1A, VvTFL1B and VvTFL1C genes (that belong to TFL1-like family) are associated with vegetative development and maintenance of meristem indetermination within the bud [73,74,77,78]. ”

Round 2

Reviewer 2 Report

Review agriculture-1079650-ver2

The article was revised and completed according to my notices; thereby, I have satisfied the 2nd version.

Author Response

Following the Editor’s comments, which we appreciate, appropriated changes have been introduced to the manuscript. We do believe that the overall quality of the manuscript has been improved.
